# ACE2-Inhibitory Effects of Bromelain and Ficin in Colon Cancer Cells

**DOI:** 10.3390/medicina59020301

**Published:** 2023-02-06

**Authors:** Babak Pakbin, Shaghayegh Pishkhan Dibazar, Samaneh Allahyari, Hanifeh Shariatifar, Wolfram Manuel Brück, Alireza Farasat

**Affiliations:** 1Werner Siemens Chair of Synthetic Biotechnology, Department of Chemistry, Technical University of Munich (TUM), Lichtenberg Str. 4, 85748 Garching bei München, Germany; 2Cellular and Molecular Research Center, Research Institute for Prevention of Non-Communicable Diseases, Qazvin University of Medical Sciences, Qazvin 34199-15315, Iran; 3Food and Drug Laboratory Research Center, Food and Drug Administration, MOH & ME, Tehran 1435643314, Iran; 4Institute for Life Technologies, University of Applied Sciences Western Switzerland Valais-Wallis, 1950 Sion, Switzerland; 5Monoclonal Antibody Research Center, Avicenna Research Institute, ACECR, Tehran 1435643314, Iran

**Keywords:** bromelain, ficin, ACE2-inhibitory, caco-2 cell line

## Abstract

*Background and Objectives*: Bromelain and ficin are aqueous extracts from fruits of *Ananas comosus* and *Ficus carcia* plants, used widely for medical applications. Angiotensin-converting enzyme 2 (ACE2) is a homolog of ACE, degrading Ang II to angiotensin 1-7 and decreasing the cellular concentration of Ang II. *Materials and Methods*: In this study, we investigated the ACE2-inhibitory, antiproliferative, and apoptosis-inducing effects of ficin and bromelain on caco-2 cells. *Results*: We found that bromelain and ficin significantly reduced the viability of human colon cancer cells with IC_50_ value concentrations of 8.8 and 4.2 mg/mL for bromelain after 24 and 48 h treatments, and 8.8 and 4.2 mg/mL for ficin after 24 and 48 h treatments, respectively. The apoptosis of the caco-2 cell line treated with bromelain was 81.04% and 56.70%, observed after 24 and 48 h. Total apoptotic proportions in caco-2 cells treated with ficin after 24 and 48 h were 83.7% and 73.0%. An amount of 1.6 mg/mL of bromelain and ficin treatments on caco-2 cells after 24 h revealed a higher decrease than that of other concentrations in the expression of ACE2 protein. *Conclusions*: In conclusion, bromelain and ficin can dose-dependently decrease the expression of ACE2 protein in caco-2 cells.

## 1. Introduction

Angiotensin-converting enzyme (ACE) is an effective and major protein risk factor causing vasoconstriction via the conversion of angiotensin I (Ang I) into angiotensin II (Ang II), contributing to the metabolism of bradykinin. Angiotensin-converting enzyme 2 (ACE2) is a homolog of ACE, degrading Ang II to angiotensin 1-7 and decreasing the cellular concentration of Ang II [1]. Ang II is also the most bioactive peptide in the renin–angiotensin system, widely participating in the progression of several chronic and acute diseases such as heart failure, hypertension, and myocardial infarction. However, ACE2 can reduce the concentration of Ang II and antagonizes these effects [2]. Lung and gastrointestinal (GI) epithelial cells have relatively high renin–angiotensin activity and are the leading sites of Ang II synthesis. Consequently, the concentrations of ACE2 are also significant in these cells [3,4].

The novel coronavirus pandemic that started at the end of 2019 (COVID-19) is a severe upper respiratory tract syndrome, caused by a coronavirus (SARS-CoV-2). Patients with COVID-19 ordinarily present with respiratory symptoms. However, several cases proved that GI symptoms also can occur in this disease [5,6]. COVID-19 infection is initiated by the invasion of the virus into the host cells in the respiratory and GI systems through viral attachment to ACE2. Consequently, ACE2 has been regarded as one of the main key receptors in SARS-CoV-2 entry into the epithelial host cells and plays an important role in COVID-19 pathogenesis [7]. As was previously mentioned, ACE2 is expressed in several different tissues in the respiratory and GI systems. However, ACE2 is expressed in other tissues such as heart, kidney, testicular, and gall bladder tissues [8]. ACE2 serves as one of the main anchors for specific domains on the spikes of SARS-CoV-2. Moreover, ACE2 plays an important role in the homeostasis of tissue inflammation through the modulation of the renin–angiotensin system [9]. It is worthwhile to note that some specific enzymatic compounds with proteolytic activity can efficiently target ACE2 and dramatically suppress the receptor activity of this macromolecule. Using these proteolytic enzymes may provide an opportunity for the effective treatment or prevention of COVID-19 [10,11].

Nowadays, several natural and synthetic types of ACE inhibitors have widely been used as the first-line treatment for different cardiovascular diseases such as heart failure, heart attack, and hypertension in humans. These ACE inhibitors include different types of proteinases [12]. Regarding the functional moiety, proteinase ACE inhibitors are divided into thiol, carboxyl, and phosphate proteolytic enzyme classes [13]. Bromelain (*Ananas comosus*) is a plant cysteine protease (thiol proteolytic enzyme class) extracted from pineapple and is currently being used as a drug for the systematic treatment of malignant, blood coagulation, and inflammatory diseases. It has also been used for complementary therapy [14,15]. It has recently been shown that bromelain induces reduced ACE2 and Transmembrane Serine Protease 2 (TMPRSS2) in VeroE6 cells. Therefore, bromelain has the potential to inhibit SARS-CoV-2 infection by targeting ACE2, TMPRSS2, and spike proteins in different cells [11]. Ficin is another plant-based sulfhydryl proteolytic enzyme belonging to the thiol protease class and elaborated by the fig tree (*Ficus carica*). Recently, ficin has been used as a complementary cancer therapy and antimicrobial agent against *Staphylococcus aureus* in some clinical trial studies [16]. It is worthwhile to note that the molecular structure of ficin presents a high sequence similarity to bromelain [17]. Ficin and bromelain can be considered the main potential treatments for SARS-CoV-2 infections. There are limited studies that have used these natural sulfhydryl proteolytic enzymes to inhibit ACE2 in human cells and treat infections caused by SARS-CoV-2. However, there are no studies available to investigate the ACE2-inhibitory activity of ficin and bromelain in GI epithelial cells. Colorectal cancer cells, including caco-2 cells, have usually been used in in vitro studies to evaluate the interactions between different compounds and human intestinal cells [18]. Therefore, the aim of this study was to determine the effects of ficin and bromelain on the suppression of ACE2 expression, cell proliferation, and inducing apoptosis in caco-2 cells.

## 2. Materials and Methods

### 2.1. Cell Culture and Treatments

All cell lines used in this assay were purchased from the Pasteur Institute cell bank (Pasteur In., Tehran, Iran) and used in this study. All cells were activated and grown in RPMI 1640 supplemented with antibiotics including penicillin (100 µL/mL) and streptomycin (100 µL/mL) (PenStrep, Gibco-Invitrogen, Carlsbad, CA, USA) and 10% (*v/v*) fetal bovine serum (FBS, Gibco-Invitrogen, Carlsbad, CA, USA), incubated with 5% CO_2_ at 37 °C and then stocked for the subsequent experiments. Cell subcultures were prepared by passaging the stocked cells into the 96-well microplates at 80% confluence after incubation with 5% CO_2_ for 48 h at 37 °C. Bromelain (Catalogue number: B5144-100UN, EC 3.4.22.32, Sigma-Aldrich, St. Louis, MO, USA) and ficin (Catalogue number: F6008-100UN, EC 3.4.22.3, Sigma-Aldrich, St. Louis, MO, USA) were purchased and used in the present study. When cells reached confluency, they were treated with various concentrations of ficin and bromelain (1, 2, 4, 8, 16, and 32 mg/mL) as the main treatments and just with the same volume of dimethyl sulfoxide (DMSO) as the negative control. Treated and negative control cells were harvested after 24 and 48 h for cell viability, apoptosis, and cell-ELISA assays. The pH value of RPMI 1640 cell culture medium was measured and adjusted to pH = 7.4 by a digital pH meter and 1 N NaOH, respectively.

### 2.2. Cell Viability Assay

The cell viability of caco-2 cells treated with bromelain and ficin enzymes was assessed by MTT assay using 3-(4,5-dimethylthiazol-2-yl)-2,5 diphenyl tetrazolium bromide (MTT) after 24 and 48 h treatments. Medium cultures in the microplates were renewed with RPMI 1640 medium culture containing 0.5 mg/mL MTT and then incubated with 5% CO_2_ at 37 °C for 4 h. Medium culture was discarded from the 96-well microplates after the incubation and DMSO was added into each well to dissolve the formazan crystals. The yellow color of MTT changed to the purple color of formazan (absorbance at 570 nm) in viable cells due to the reduction of MTT to formazan. To evaluate the viable cells in each well, the color change was measured using an ELISA microplate reader model Elx808 (BioTek, Winooski, VT, USA). The cell viability percentage was calculated using the following formula:Cell viability (%) = Ae − An/Ac − An × 100
where Ae, An, and Ac represent the absorbance of the experiment, the absorbance of the blank, and the absorbance of the control, respectively. Fifty percent inhibitory concentrations (IC_50_ values) of bromelain and ficin on caco-2 cells were determined while the treatments decreased the cell viability to 50%. The IC_50_ concentrations of bromelain and ficin treatments were considered to treat the caco-2 cells in apoptosis analysis [18].

### 2.3. Cell Apoptosis Analysis

Apoptosis in caco-2 cancer cells induced by bromelain and ficin treatments was evaluated by using flow cytometry. A FACS-Calibur flow cytometer (Dickinson Immunocytometry system, CA, USA) and a commercial apoptosis kit (eBioscience, San Diego, CA, USA) containing Annexin V-FITC and propidium iodide (PI) were used according to the manufacturers’ instructions to analyze apoptosis in the treated cells. A total of 106 cells were seeded in each well, treated with bromelain and ficin (DMSO as the negative control), and then incubated with 5% CO_2_ at 37 °C for 24 and 48 h. All cells were harvested, washed twice with phosphate-buffered saline (PBS), and then incubated at room temperature in the dark with Annexin V-FITC and PI for 30 and 5 min, respectively. The expression of Annexin V-FITC and PI staining was documented by using flow cytometry [19].

### 2.4. Cell-ELISA Method

The reactivity of anti-ACE2 was scrutinized using standard caco-2 ACE2-positive and ACE2-negative cell lines (Pasteure institute, Tehran, Iran). Cells were cultured in 96-well culture microplates in RPMI 1640 supplemented with antibiotics including penicillin (100 µL/mL) and streptomycin (100 µL/mL) (PenStrep, Gibco-Invitrogen, Carlsbad, CA, USA) and 10% (*v/v*) fetal bovine serum (FBS, Gibco-Invitrogen, Carlsbad, CA, USA). Adhered caco-2 cells were fixed with 5% *v/v* formaldehyde, and the supernatants were removed. Formaldehyde was discarded after 30 min of incubation at room temperature. Then, all wells were washed three times with PBS, blocked with 5% bovine serum albumin (BSA, Gibco-Invitrogen, Carlsbad, CA, USA), and incubated at 37 °C for 2 h. Purified anti-ACE2 (0.5 μg/well) was added into each well and incubated at 4 °C for 75 min. Then, all wells were washed in triplicate with washing buffer containing PBS and 1% *v/v* BSA and incubated with 100 μL of anti-his-tag-HRP (1:5000 diluted) on ice for 1 h followed by five washes with the buffer to reduce the background. To determine the relative quantity of bound anti-ACE2, 3,3′,5,5′-Tetramethylbenzidine (TMB, Sigma-Aldrich, St. Louis, MO, USA) was added. The developed color in each well was evaluated at 450 nm by using an ELISA microplate reader model (Elx808, BioTek, Winooski, VT, USA) [20].

### 2.5. Statistical Analysis

One-way analysis of variance (ANOVA) was applied to determine the significant (*p* < 0.05) levels of difference among the different groups of data by using SPSS version 23.0.0 (SPSS Inc., Chicago, IL, USA). All measurements were performed in triplicate.

## 3. Results

### 3.1. Antiproliferation Assay (MTT)

The antiproliferative effects of bromelain and ficin against colon cancer cells were assessed using the cell viability of caco-2 cancer cells and determined using the MTT assay. Figure 1 and Figure 2 show the percentage of viable caco-2 cells treated with bromelain and ficin after 24 and 48 h treatments. Bromelain significantly (*p* < 0.05) reduced caco-2 cell viability after 24 and 48 h in concentrations of 16 and 32 mg/mL. However, ficin was able to reduce the cell viability of caco-2 cancer cells significantly (*p* < 0.05) more at a concentration of 32 mg/mL than at any other concentration. The reduction in caco-2 cell viability treated with both bromelain and ficin was found to be dose-dependent. As we found the antiproliferative activity of bromelain and ficin enzymes against colon cancer cells in this study, we were further encouraged to investigate apoptosis in the treated cancer cells. To evaluate and analyze the apoptosis induction, bromelain and ficin treatments with IC_50_ concentrations were regarded after 24 and 48 for caco-2 cancer cells. The IC_50_ concentrations of bromelain against caco-2 cancer cells were measured at 8.8 and 4.2 mg/mL for the 24 and 48 h treatments, respectively. Moreover, the IC_50_ value concentrations of ficin against caco-2 cells were calculated at 15 and 15.5 mg/mL after 24 and 48 h, respectively.

### 3.2. Apoptosis Analysis

Apoptosis was measured by flow cytometry to determine the cell viability of caco-2 cells treated by the bromelain and ficin enzymes. Figure 3 and Figure 4 demonstrate the apoptosis-inductive effects of bromelain and ficin enzymes, respectively, with IC_50_ concentrations after 24 and 48 h treatments, against caco-2 cancer cells. The Q1, Q2, Q3, and Q4 quadrants show the proportions of necrosis, late apoptosis, early apoptosis, and viable cells, respectively, in each figure. The proportions of late (Q2) and early (Q3) apoptosis were added to each other to determine the total proportion of apoptotic cells. As has already been visible in Figure 3 and Figure 4, the proportion of total apoptotic cells in the caco-2 cell line control sample (just treated with DMSO) was calculated to be 18.98%. In caco-2 cells treated with bromelain, however, it was observed to be 81.04% and 56.70% after 24 and 48 h, respectively. Figure 4 illustrated 83.7% and 73.0% apoptosis in caco-2 cancer cells treated with ficin after 24 and 48 h, respectively. Ficin contributed significantly (*p* < 0.05) more than bromelain to apoptosis induction in caco-2 cancer cells. It is worthwhile to note that the apoptosis induction caused by both bromelain and ficin enzymes in caco-2 cells was significantly (*p* < 0.05) increased after 24 and 48 h treatment in comparison with the control samples (treated with DMSO).

### 3.3. Cell-ELISA Method

In this study, the expression of ACE2 protein in caco-2 cells treated with bromelain and ficin after 24 and 48 h treatments was assessed using the cell-ELISA method. Absorbance at 450 nm showed each well’s relative quantity of bound anti-ACE2 and 3,3′,5,5′-Tetramethylbenzidine. Figure 5 and Figure 6 demonstrate the absorbance at 450 nm for caco-2 cells treated with bromelain and ficin enzymes, respectively. As can be seen in Figure 5, 1.6 mg/mL of bromelain treatment on caco-2 cells after both 24 and 48 h treatments revealed a significantly (*p* < 0.05) higher decrease than that of other concentrations in absorbance at 450 nm in the cell-ELISA assay and the expression of ACE2 protein. Treatment with ficin at a concentration of 1.6 mg/mL also indicated a significantly (*p* < 0.05) higher decrease than that of other concentrations in the expression of ACE2 protein in caco-2 cells after 24 h treatment. However, the significantly (*p* < 0.05) highest decrease in the expression of ACE2 protein in caco-2 cells treated with ficin after 48 h treatment was observed at a concentration of 0.8 mg/mL in comparison with the other concentrations. The results also showed that bromelain and ficin enzymes could dose-dependently decrease the expression of ACE2 protein in caco-2 cells.

## 4. Discussion

Bromelain is an aqueous extract from the *Ananas comosus* plant’s immature fruits containing a complex mixture of various thiol-peptidases and other components such as cellulase, peroxidase, phosphatase, and glucosidase [21]. Bromelain has previously been used in several in vitro and animal studies for a different range of medical practices including preventing edema formation, reducing blood fibrinogen levels, supporting fibrinolysis, plasmin activation, preventing the aggregation of blood platelets, reducing blood plasma-kinins levels, preventing the adhesion of blood vessel epithelial cells to platelets, inducing the secretion of proinflammatory factors in tumor cells, supporting the oxidative burst, and acting as an anti-inflammatory agent [22,23]. In this study, we observed antiproliferative and anticancer activities against colon cancer cells with IC_50_ value concentrations of 8.8 and 4.2 mg/mL after 24 and 48 h treatments, respectively. We observed a dose-dependent anti-proliferative effect of bromelain against caco-2 cells. We also found that 81.04% and 56.70% of caco-2 cancer cells treated with bromelain were in apoptotic stages after 24 and 48 h treatment, respectively. Müller et al. (2016) showed the antitumor and cytotoxic effects of bromelain against the human cholangiocarcinoma cell lines [24]. Raeisi et al. (2019) also investigated the inhibitory potential of bromelain against the proliferation and colony formation of human stomach, breast, and prostate cancer cells. They observed the potent antitumor activities of bromelain against cancer cells and determined IC_50_ concentrations of 65, 60, and 65 µg/mL to treat human stomach, prostate, and breast cancer cells, respectively. They also found that bromelain has a dose-dependent anti-proliferative activity against cancer cells [25].

*Ficus carcia* belongs to the Moraceae family, including more than 700 species growing in tropical areas around the world [26]. *F. carcia* fruits are commonly used for their human health-promoting characteristics. However, their leaves and latex have been consumed for medical treatment strategies in different health situations such as warts, calluses, gum wounds, bee stings, and cataract treatments [27]. Ficin, a complex mixture of cysteine proteases, is extracted from *F. carcia* latex. Ficin has demonstrated several medical applications, such as apoptosis in cancer cells and inhibitory effects on bacterial pathogen biofilm formation [28]. In the present study, we also demonstrated the anti-proliferative potential of ficin against human colon cancer cells. We also observed the dose-dependent antitumor and apoptosis-inducing effects of ficin against caco-2 cancer cells. Hashemi et al. (2011) demonstrated that *F. carcia* latex could inhibit the proliferation of stomach cancer cells without any cytotoxicity against normal human cells [29]. Shin et al. (2017) also showed the efficient anti-proliferative and apoptosis-inducing effects of *F. carcia* latex against human hypopharynx squamous carcinoma cells [30].

COVID-19 is caused by SARS-CoV-2. The renin–angiotensin–aldosterone system (RAAS) is one of the most important factors in SARS-CoV-2 pathogenesis in pulmonary, cardiovascular, kidney, and gastric physiology. ACE2 is one of the principal components of the RAAS, identified as the main SARS-CoV-2 binding site on pulmonary, cardiovascular, kidney, and gastric cells and mediating the viral entry for SARS-CoV-2 [31]. RAAS mediates two principal biological pathways: (1) converting Ang I into Ang II (acting at the Ang II type 1 receptor contributing to blood pressure increases through renal water, vasoconstriction, and sodium reabsorption increase and endothelial dysfunction via the stimulation of proinflammatory cytokines leading to inflammation promotion); (2) the utilization of ACE2 to generate Ang 1-7 (acting at the Mas receptor, contributing to inflammation and blood pressure reduction) [32]. ACE2 mediates the metabolization of Ang II to Ang 1-7 and converts Ang I into Ang 1-9. Due to the interaction between ACE2 protein and SARS-CoV-2, viral propagation induced by the expression of RAAS-inhibitor-induced ACE2 has been regarded as an important mechanism associated with cardiovascular, hypertensive, and chronic kidney diseases, COVID-19 severity, and SARS-CoV-2 infection in different tissues [33,34]. Regarding this theory, any significant decreasing changes caused by different compounds on the expression of ACE2 protein and its effect on the infection of SARS-CoV-2-affected cells can be considered a practical and efficient strategy to treat or prevent COVID-19 [35].

In this study, we found that bromelain and ficin could suppress the expression of ACE2 protein in human colon cancer cells in a dose-dependent manner. Kritis et al. (2020) investigated the effects of a bromelain and curcumin combination as an immune-boosting nutraceutical agent against severe cases of COVID-19. They reported that bromelain inhibits the infection caused by SARS-CoV-2 via the downregulation of ACE2 protein expression [36]. Sagar et al. (2021) demonstrated that bromelain treatment dramatically diminishes the expression of ACE2 protein in VeroE6 cells [10]. Akhter et al. (2021) also demonstrated that the combination of acetylcysteine and bromelain could synergistically suppress the expression of human host cells infected with SARS-CoV-2 and efficiently prevent COVID-19 [37]. The overall results obtained in our present study suggest that bromelain and ficin could also be used to potentially treat SARS-CoV-2 infections in different tissues and in severe cases of COVID-19. We also observed the greater ACE2-inhibitory activity of bromelain and ficin than that of some drugs such as Leu-Ser-Gly-Tyr-Gly-Pro (LSGYGP) peptide and tilapia skin gelatin hydrolysates in caco-2 cells, and we found these cysteine proteases more efficient than some other ACE2 inhibitor drugs [38,39].

## 5. Conclusions

Ficin and bromelain are cysteine proteases known as ACE inhibitors. In this study, we showed that bromelain and ficin efficiently decreased proliferation in the caco-2 cell line dose-dependently. Both ficin and bromelain also induced apoptosis in treated caco-2 cells. We also found that ficin induced significantly more apoptosis than bromelain in caco-2 cells. The expression of ACE2 was suppressed in caco-2 cells treated with both ficin and bromelain dose-dependently; however, ficin decreased the expression of ACE2 significantly more than bromelain in caco-2 cells. Consequently, ficin showed more ACE2-inhibitory activity than bromelain; however, we observed this activity to be greater than that of some drugs such as LSGYGP peptide and tilapia skin gelatin hydrolysates.

## Figures and Tables

**Figure 1 medicina-59-00301-f001:**
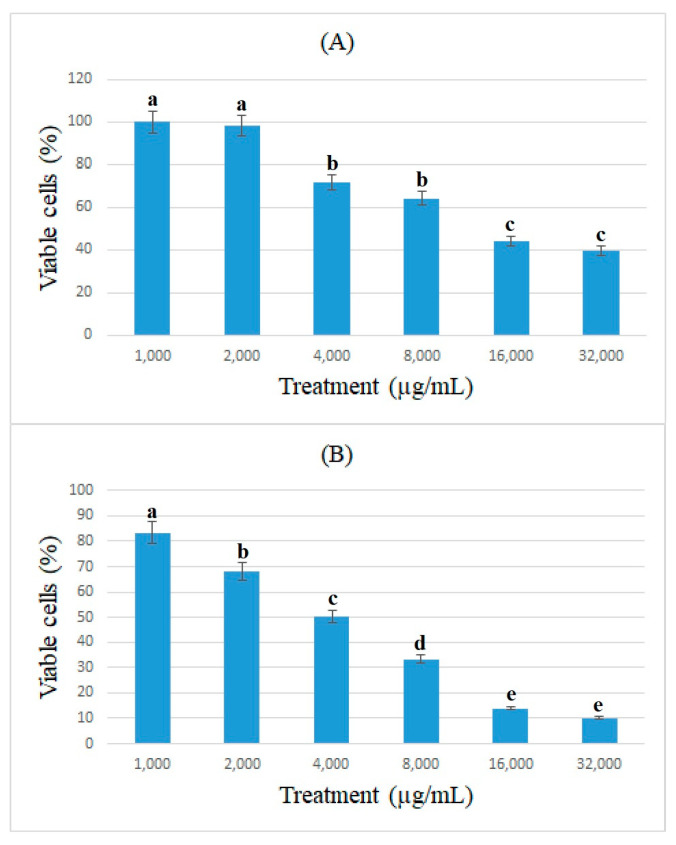
Cell viability of caco-2 cells under exposure to different concentrations of bromelain evaluated by MTT assay after 24 (**A**) and 48 h (**B**) treatments. Alphabetical letters indicate significant differences.

**Figure 2 medicina-59-00301-f002:**
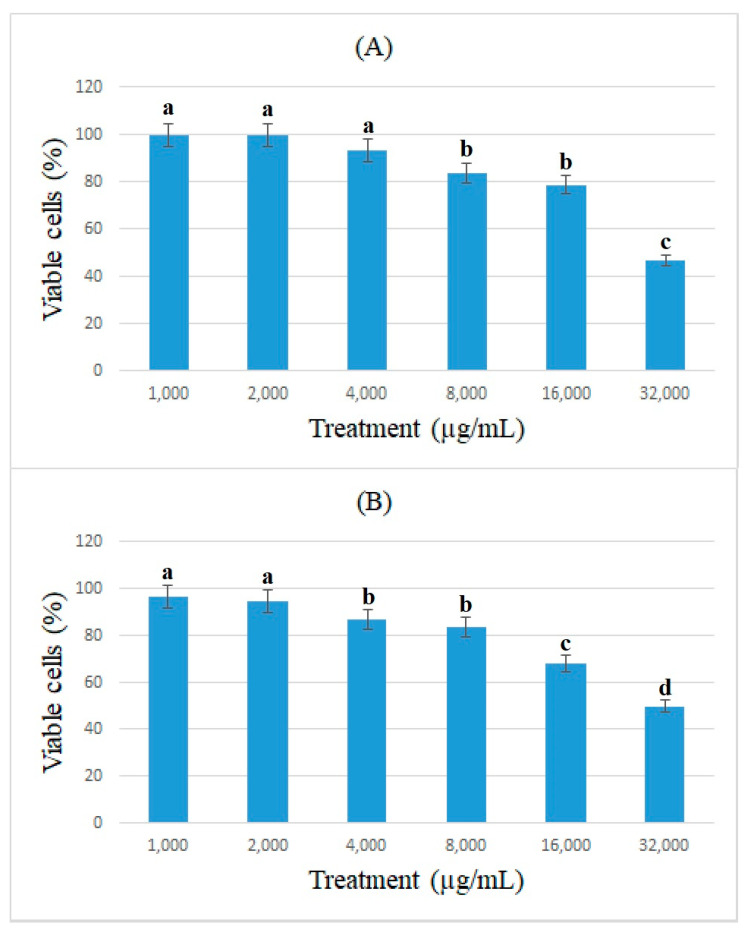
Cell viability of caco-2 cells under exposure to different concentrations of ficin evaluated by MTT assay after 24 (**A**) and 48 h (**B**) treatments. Alphabetical letters indicate significant differences.

**Figure 3 medicina-59-00301-f003:**
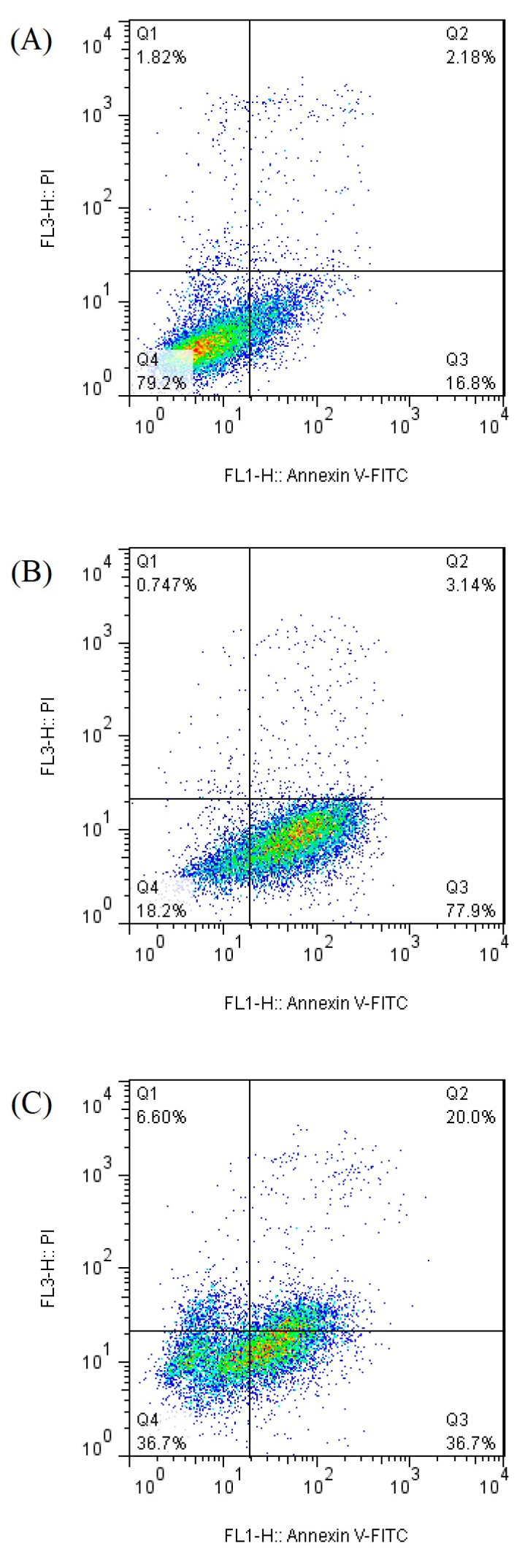
Apoptotic analysis of caco-2 cells treated with bromelain (negative control samples (**A**), after 24 h (**B**), and after 48 h (**C**) of treatment) by using the flow cytometry method.

**Figure 4 medicina-59-00301-f004:**
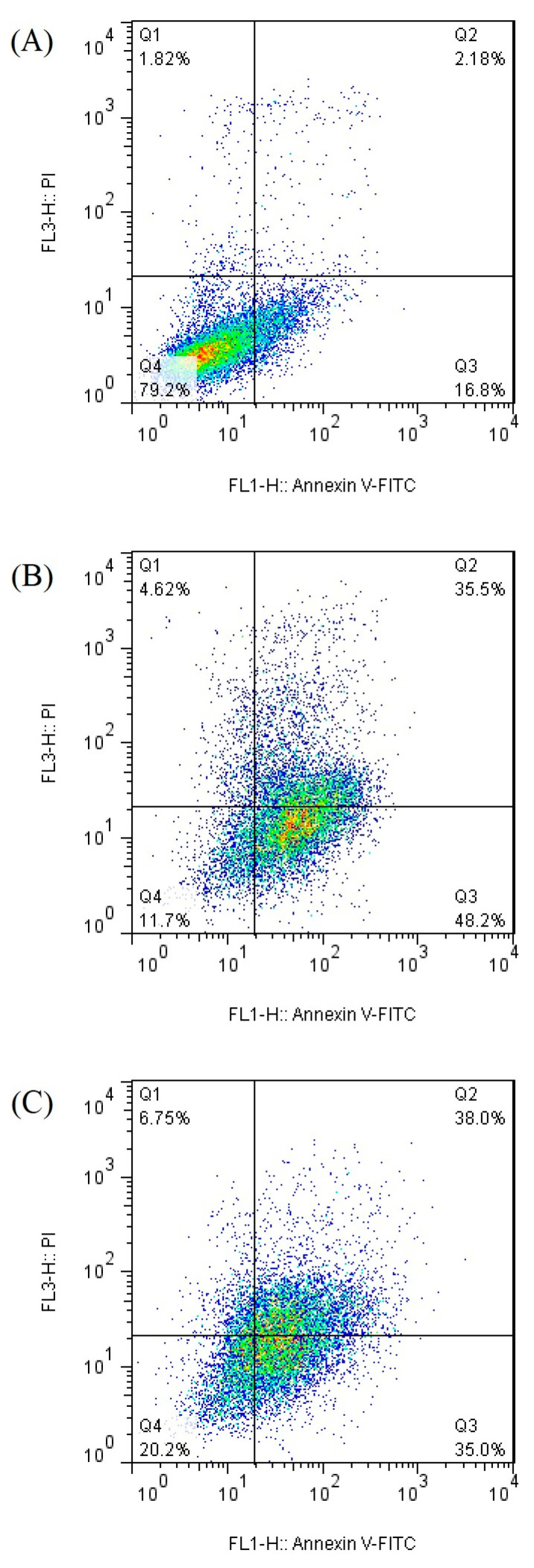
Apoptotic analysis of caco-2 cells treated with ficin (negative control samples (**A**), after 24 h (**B**), and after 48 h (**C**) of treatment) by using the flow cytometry method.

**Figure 5 medicina-59-00301-f005:**
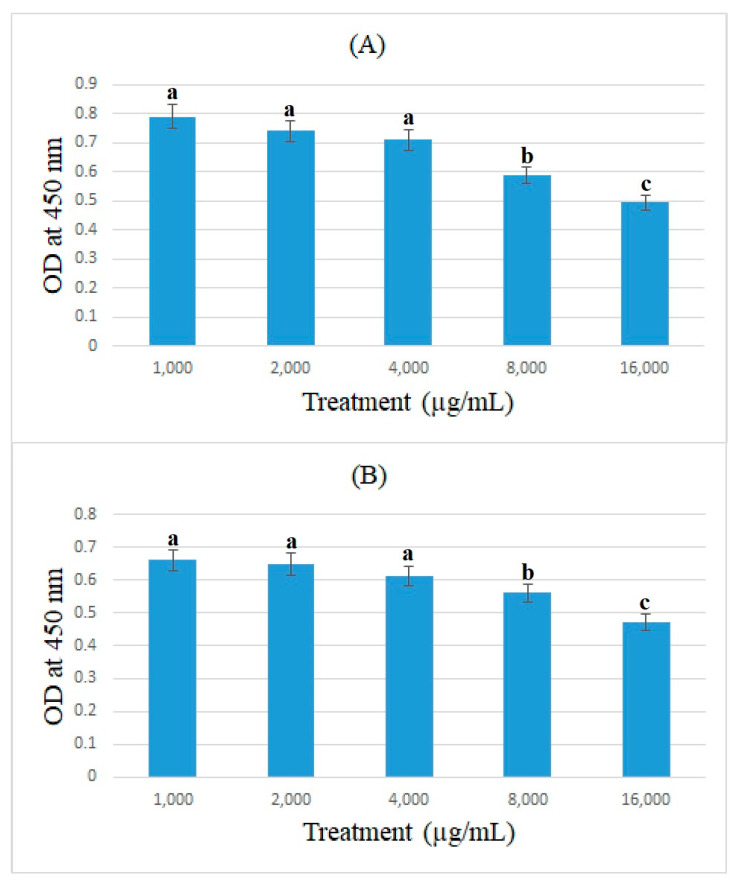
ACE2 concentrations (optical density (OD) at 450 nm) in caco-2 cells under exposure to different concentrations of bromelain evaluated by MTT assay after 24 (**A**) and 48 h (**B**) treatments. Alphabetical letters indicate significant differences.

**Figure 6 medicina-59-00301-f006:**
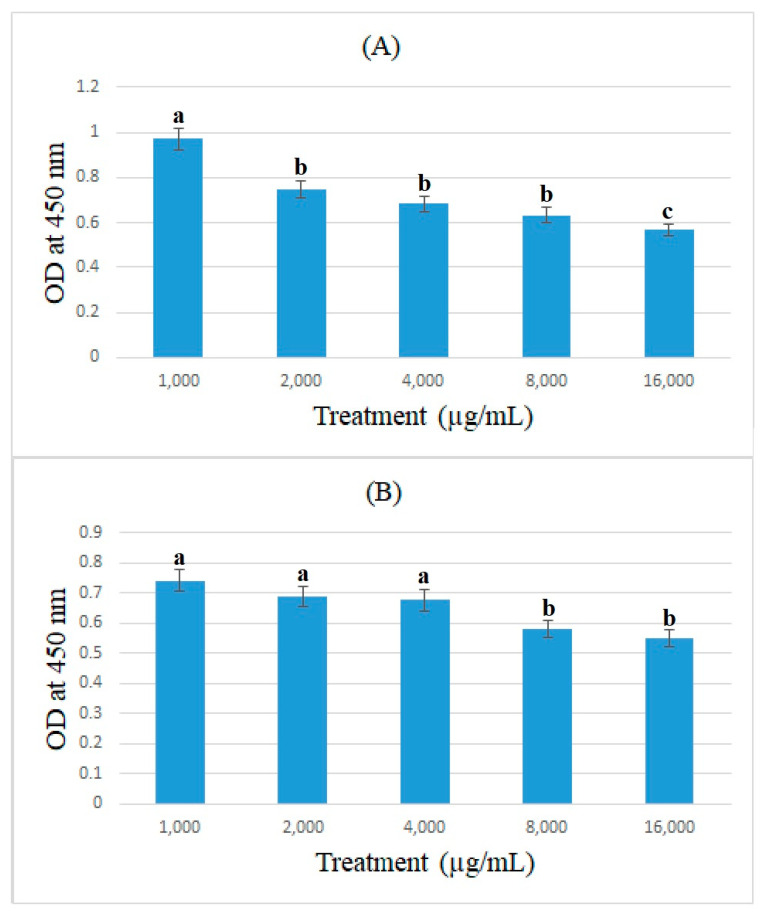
ACE2 concentrations (OD at 450 nm) in caco-2 cells under exposure to different concentrations of ficin evaluated by MTT assay after 24 (**A**) and 48 h (**B**) treatments. Alphabetical letters indicate significant differences.

## Data Availability

We confirm that all data and findings of this study are available within the article.

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
