# Peer review of "ACE2-Inhibitory Effects of Bromelain and Ficin in Colon Cancer Cells"

_medicina, 2023, doi:10.3390/medicina59020301_

Round 1
Reviewer 1 Report (Previous Reviewer 1)
The authors have made a number of corrections, according to my previous comments.
However, some shortcomings still remain:
1. The authors do not compare the efficiency of bromelain and ficin with the efficiency of already published drugs (according to the literature). This should be done in section 4. Discussion or 5. Conclusion
2. Conclusion is too short. It should be expanded, we will add information about the possible continuation of research, about their use in practice, etc.
3. The authors write that ficin and bromelain are serine proteases, but they are cysteine proteases. It needs to be fixed
Author Response
Dear Reviewer 1,
Thank you very much for your precise reviewing, practical and helpful comments. All revisions have been addressed and implemented successfully according to your comments as described below:
- The efficiency of bromelain and ficin is compared with some other drugs used to inhibit the ACE2 expression in caco-2 cells and it is added into Discussion and Conclusion sections. Also, the references are revised accordingly.
- Conclusion section of the manuscript is developed and some more information is added according to your comments.
- The serine protease is revised to cysteine protease in the manuscript.

Reviewer 2 Report (Previous Reviewer 2)
After reviewing the entire contribution, and given the length of the article and the depth that has been given to the studies, I recommend to the authors and the editor to change the format of this contribution to "short communication". This comment has previously been directed to the editor.
Author Response
Dear Reviewer 2,
Thank you very much for your precise reviewing, practical and helpful comments. Some revisions have been done in the manuscript. Also, the format has been changed to Communication for this manuscript according to your comment.

This manuscript is a resubmission of an earlier submission. The following is a list of the peer review reports and author responses from that submission.
Round 1
Reviewer 1 Report
The article «ACE2 inhibitory effects of Bromelain and Ficin in Colon Cancer Cells» has a high applied value.
The present article demonstrated the effects of ficin and bromelain on viability, apoptosis inducing and suppression of ACE2 protein expression in caco-2 cells.
Authors found that bromelain and ficin can dose-dependently decrease the expression of ACE2 protein in caco-2 cells.
The article is written logically and clearly. However, the text of article needs significant revision.
Major comments:
1. Phrase in Abstract: “In this study, we demonstrated the effects of ficin and bromelain on via-bility, apoptosis inducing and suppression of ACE2 protein expression in caco-2 cells”. This is results rather than methods.
2. The authors did not directly study SARS-CoV-2 entry into the host cells. I think that discussions about COVID19 should be removed at least from the Abstract and Conclusion, and this part should be shortened in the introduction. Bromelain as potential efficient treatments of COVID19 known for a long time (https://link.springer.com/book/10.1007/978-3-319-28570-2). However, the authors do not compare the efficiency of their enzymes with the efficiency of already published proteases.
Minor comments:
Line 75: Staphylococcus aureus should be italicized
Line 125: CO2, 2 should be subscripted
The manuscript is poorly formatted: many half-blank pages
Line 228: in-vitro should be italicized
Lines 247, 251 and 256: F. carcia should be italicized
Author Response
Dear reviewer 1,
Thank you very much for your valuable comments. All revisions have been implemented and addressed throughout the manuscript, in the track-change version.
Major comments,
- The phrases in the abstract section are revised according to your comment.
- All COVID-19 issues in the abstract and conclusion sections are removed and revised according to your comments.
Minor comments
- Staphylococcus aureus is italicized.
- CO2, 2 is subscripted.
- Half-blank pages are revised throughout the manuscript.
- in-vitro is italicized.
- carcia names are italicized.
Reviewer 2 Report
This work deals with bromelain and ficin, two aqueous extracts of fruits, and their antitumor effects on CRC cells. In addition, the negative modulation of a key regulator of the renin-angiotensin pathway was also evaluated.
This work is too short to be considered as a research article. From the experimental section, too few approaches are valued.
The objective of the study is unclear. As it reads, is to prove that both natural compounds, Bromelain and Ficin, can reduce cell viability and to induce apoptosis in intestinal epithelial cells infected with Sars-cov-2. How would this benefit a person who is already infected with the virus? On the other hand, if the cell is already infected, the virus has already entered via host cell receptor-envelope protein binding. At this point, what is the benefit of reducing ACE2 expression? this should be further discussed-
On the other hand, CRC tumor cells are used for this study, what is the level of ACE2 expression in CRC cells compared to cells or normal intestinal tissue?
I would recommend the authors to redefine the objective of the work. For example: lean towards studies that prove antiviral activity (and with implementation of assays with viruses) or investigate the antitumor capacity of the compunds, and really focus on one of the objectives without mixing two pathologies since it does not apply to these studies. On the other hand, CRC is a risk-based disease for COVID-19 patients; this has not been addressed or mentioned by the authors.
It is not explicit how the transgenic cell lines lacking or with the ACE2 gene were obtained.
The Bromelain from pineapple compund that I found on Sigma aldrich page (Merck) says uncertain concentration of the compound (catalog number #B4882, ≥3 units/mg protein- #B51445-15 units/mg protein). Similar information was found when consulting about ficin. Therefore, if the authors used these compounds, how did they calculate the dose?
Also, with those high doses of both tested compounds, what was the final pH in the culture medium? The observed effects were attributable only to the treatmens or there is an important effect on the pH, maybe acidification of the media? I read in methodology that the cells were cultured in RPMI, and phenol red (pH indicator) is not mentioned.
Author Response
Dear Reviewer 2,
Thank you very much for your prestigious and helpful comments. All revisions you considered have been implemented and addressed throughout the manuscript, in the track-change version.
- The objective of the study is the ACE2 inhibitory, antiproliferative and inducing apoptosis effects of ficin and bromelain on caco-2 cells. This statement is stated throughout the manuscript and other sections are revised according to the aim the study.
- CRC cells are used as human intestinal cells, which were also previously used by other researchers as representative of human intestinal cells, to evaluate ACE2 inhibitory, antiproliferative and inducing apoptosis effects. This statement is added into the manuscript.
- We used standard ACE2 positive and negative caco-2 cells purchased from Pasteur institute. This statement is added into the manuscript.
- We used enzymes with definite concentrations. The catalogue number including the concentrations of each enzyme are added into the methods section of the manuscript.
- The pH value of the RPMI medium was measured and adjusted before any treatment. This statement is added into the methods section.
Round 2
Reviewer 2 Report
no further comments